# Added Value of Viscoelasticity for MRI-Based Prediction of Ki-67 Expression of Hepatocellular Carcinoma Using a Deep Learning Combined Radiomics (DLCR) Model

**DOI:** 10.3390/cancers14112575

**Published:** 2022-05-24

**Authors:** Xumei Hu, Jiahao Zhou, Yan Li, Yikun Wang, Jing Guo, Ingolf Sack, Weibo Chen, Fuhua Yan, Ruokun Li, Chengyan Wang

**Affiliations:** 1Human Phenome Institute, Fudan University, Shanghai 201203, China; 21212030009@m.fudan.edu.cn; 2Department of Radiology, Ruijin Hospital, School of Medicine, Shanghai Jiao Tong University, Shanghai 200025, China; zjh01e31@rjh.com.cn (J.Z.); ly40730@rjh.com.cn (Y.L.); wyk01g67@rjh.com.cn (Y.W.); yfh11655@rjh.com.cn (F.Y.); 3Department of Radiology, Charité–Universitätsmedizin Berlin, 10117 Berlin, Germany; jing.guo@charite.de (J.G.); ingolf.sack@charite.de (I.S.); 4Philips Healthcare, Shanghai 200070, China; weibo.chen@philips.com

**Keywords:** Ki-67, hepatocellular carcinoma (HCC), conventional MRI (cMRI), magnetic resonance elastography (MRE), deep learning combined radiomics (DLCR)

## Abstract

**Simple Summary:**

This study aimed to explore the added value of magnetic resonance elastography (MRE) in the prediction of Ki-67 expression in hepatocellular carcinoma (HCC) using a deep learning combined radiomics (DLCR) model. A total of 108 histopathology-proven HCC patients who underwent preoperative MRI and MR elastography were included. All the patients were divided into training and validation cohorts. An independent cohort including 43 patients was included for testing. A DLCR model was proposed to predict the expression of Ki-67 with conventional MRI (cMRI) as inputs. The images of shear wave speed (*c*-map) and phase angle (*φ*-map) derived from MRE were also fed into the DLCR model. Experimental results show that both *c* and *φ* values were ranked within the top six features for Ki-67 prediction with random forest selection, which revealed the value of MRE-based viscosity for the assessment of the tumor proliferation status in HCC. The model with all modalities (MRE, AFP, and cMRI) as inputs achieved the highest AUC of 0.90 ± 0.03 (CI: 0.89–0.91) in the validation cohort. The same finding was observed in the independent testing cohort with an AUC of 0.83 ± 0.03 (CI: 0.82–0.84). MRE-based *c* and *φ*-maps can serve as important parameters to assess the tumor proliferation status in HCC.

**Abstract:**

This study aimed to explore the added value of viscoelasticity measured by magnetic resonance elastography (MRE) in the prediction of Ki-67 expression in hepatocellular carcinoma (HCC) using a deep learning combined radiomics (DLCR) model. This retrospective study included 108 histopathology-proven HCC patients (93 males; age, 59.6 ± 11.0 years) who underwent preoperative MRI and MR elastography. They were divided into training (*n* = 87; 61.0 ± 9.8 years) and testing (*n* = 21; 60.6 ± 10.1 years) cohorts. An independent validation cohort including 43 patients (60.1 ± 11.3 years) was included for testing. A DLCR model was proposed to predict the expression of Ki-67 with cMRI, including T2W, DW, and dynamic contrast enhancement (DCE) images as inputs. The images of the shear wave speed (*c*-map) and phase angle (*φ*-map) derived from MRE were also fed into the DLCR model. The Ki-67 expression was classified into low and high groups with a threshold of 20%. Both *c* and *φ* values were ranked within the top six features for Ki-67 prediction with random forest selection, which revealed the value of MRE-based viscosity for the assessment of tumor proliferation status in HCC. When comparing the six CNN models, Xception showed the best performance for classifying the Ki-67 expression, with an AUC of 0.80 ± 0.03 (CI: 0.79–0.81) and accuracy of 0.77 ± 0.04 (CI: 0.76–0.78) when cMRI were fed into the model. The model with all modalities (MRE, AFP, and cMRI) as inputs achieved the highest AUC of 0.90 ± 0.03 (CI: 0.89–0.91) in the validation cohort. The same finding was observed in the independent testing cohort, with an AUC of 0.83 ± 0.03 (CI: 0.82–0.84). The shear wave speed and phase angle improved the performance of the DLCR model significantly for Ki-67 prediction, suggesting that MRE-based *c* and *φ*-maps can serve as important parameters to assess the tumor proliferation status in HCC.

## 1. Introduction

Liver cancer is the fourth most common cancer and the second leading cause of malignant death. As the most common histological type, hepatocellular carcinoma (HCC) accounts for the majority of incidence and mortality of liver cancer cases [1]. HCC is highly heterogeneous and exhibits various biological behaviors. Ki-67 is one of the most common used cell proliferation biomarkers representing tumor aggressiveness. Previous studies showed that expression of Ki-67 is valuable in selecting recipients for liver transplant [2], patients with high Ki-67 expression are often encouraged to receive preoperative adjuvant therapy to improve prognosis [3], and Ki-67-targeted therapy is an attractive and promising avenue in HCC treatment [4]. Thus, the precise evaluation of Ki-67 expression preoperatively may be beneficial to appropriate clinical decision. However, current clinical practice guidelines recommend imaging to establish diagnosis of HCC noninvasively rather than through confirmatory biopsy. Therefore, the noninvasive preoperative evaluation of Ki-67 expression in HCC is warranted to outline personalized treatment strategies in clinical practice.

With the rapid development of deep learning models and radiomics theories, they have become effective methods of predicting tumor aggressiveness [5]. Preliminary studies showed the feasibility of MRI-based radiomics or deep learning models in predicting the Ki-67 expression of HCC. Current results were mainly based on the assessment of vascularity derived from Gd-DTPA-enhanced images. Hu et al. [6] reported that histogram-based parameters from apparent diffusion coefficient (ADC) maps and arterial phase (AP) images could be used to determine the Ki-67 labeling index (LI) in HCC with an AUC up to 0.826. Similarly, Fan et al. [7] proposed a combined model including an arterial phase-based RAD-score and serum alpha-fetoprotein (AFP) level to predict Ki-67 expression preoperatively. The combined model (AUC, 0.863), that included the AP RAD-score and serum AFP level, demonstrated improved performance when compared with the single AP radiomics model (AUC, 0.813). Recently, emerging evidence confirmed that hepatobiliary agent-enhanced MR imaging might provide valuable information to characterize HCC tumor biology independent of vascularity. Li et al. [8] found that texture analysis on the hepatobiliary phase (HBP), arterial phase (AP), and portal vein phase (PVP) images were useful in predicting Ki67 LI. However, the application of the hepatobiliary agent was limited by severe hepatic dysfunction or cholestasis, the relatively weak arterial phase hyperenhancement, and the relatively high frequency of arterial phase artifacts. In addition, interpreting the complex associations between the deep learning or radiomics features and biologic processes of HCC still remains an enormous challenge.

From the view of biomechanics, MR elastography (MRE) can provide insight into the mechanisms governing liver biology. Elastography applies mechanical tension or stimuli to tissue, monitors the tissue’s response to the induced stimulus, and uses it to reconstruct parameters that characterize the mechanical properties of the image-encoded response. A variety of factors, including cell types, extracellular matrix deposition, cellularity, and fluid transport, alter the mechanical properties of biological organization [9,10]. MRE can also be applied on the liver [11,12]. MRE can measure displacement due to propagating mechanical waves, from which biomechanical properties of tumors, such as stiffness, can be calculated [13]. However, stiffness quantified by MRE was always used for depicting diffuse liver disease, such as liver inflammation, fibrosis, and portal hypertension in chronic hepatitis patients [11]. To update, nearly all elastography techniques measure stiffness without taking into account the viscoelastic, anisotropic, and nonlinear properties of most living tissues. The complex shear modulus of biological tissues is a composite dimension consisting of a storage modulus and a loss modulus, representing the elasticity and viscosity, respectively.

Tomoelastography, as an advanced MRE technique [14], yields quantitative maps of shear wave speed (*c* in m/s) and loss angle (*φ* in rad) as proxies of tissue stiffness and viscosity, respectively. These two parameters, especially viscosity, have demonstrated great potential for tumor detection and aggressiveness prediction of prostate cancer, pancreatic cancer, and liver cancer [12,15,16]. For malignant tumors, increased cellularity and aggressiveness not only leads to the higher stiffness but also changes the tumor microenvironment, which have liquid properties [17,18]. Moreover, tomoelastography provides full field-of-view (FOV) quantitative maps with spatial resolution comparable to conventional MR images, which is suitable for analysis by deep learning or radiomics models. To update, the biomechanical properties of HCC have not been investigated for aggressiveness evaluation.

Our study aimed to investigate the value of viscoelasticity derived from tomoelastography for predictive capability of Ki-67 expression in HCC using an MRI-based deep learning combined radiomics (DLCR) model.

## 2. Materials and Methods

### 2.1. Patients

This retrospective study was approved by our institutional review board and local ethics committee in Ruijin Hospital, Shanghai, China. Written informed consent was obtained from each participant. Between June 2020 and June 2021, 184 patients with suspected HCCs were initially enrolled to establish the predictive model. For patients with multiple lesions, the largest one was selected to avoid clustering effect. The inclusion criteria were as follows: (1) age ≥ 18 years; (2) patients who underwent conventional MRI and preoperative tomoelastography examinations. The exclusion criteria were: (1) poor image qualities (*n* = 3); (2) prior chemotherapy or radiotherapy (*n* = 24); (3) no histopathological results (*n* = 6) (Figure 1). Finally, 108 patients (number of males, 93; number of females, 13; age, 59.6 ± 11.0 years) were included. All the demographics and clinical characteristics of patients were recorded (Table 1). They were randomly divided into a training cohort (76 males; 11 females; age, 59.4 ± 11.1 years) and a testing cohort (17 males; 4 females; age, 60.38 ± 10.20 years) with a ratio of 8:2. Whereafter, between July 2021 and November 2021, 43 HCC patients (36 males; 7 females; age, 60.01 ± 11.10 years) who met the above criteria were included as an independent validation cohort. The median time of interval between liver resection and MRI examination was 2 days, ranging from 1 to 3 days.

### 2.2. Conventional MRI (cMRI)

Conventional MRI was performed on two 1.5 T MR scanners (Magnetom Aera, Siemens, Munich, German; uMR 660, United Imaging, Shanghai, China) and a 3.0 T MR scanner (Ingenia, Philips, Amsterdam, The Netherlands). The imaging protocol consisted of T1-weighted (T1w), T2w, diffusion weighted imaging (DWI) with b-values of 0, 50, and 800 s/mm^2^, and multiphase dynamic contrast-enhanced (DCE) imaging (Appendix A).

### 2.3. Tomoelastography

Tomoelastography was performed using a 1.5 T MR scanner (Magnetom Aera, Siemens, Germany). Continuous harmonic vibrations at frequencies of 30, 40, 50, and 60 Hz were induced by an external pressurized-air driving system. Two rear pneumatic actuators (0.6 bar) and one front pneumatic actuator (0.4 bar) were positioned near the liver. To ensure a stable mechanical state, apply vibration 5 s before imaging. Three-dimensional wavefields were acquired using a single-shot spin-echo echo-planar imaging (EPI) sequence with flow-compensated motion encoded gradients (MEG). Fifteen consecutive transverse slices of images were acquired with an FOV of 384 × 312 mm^2^ and a resolution of 3 × 3 × 5 mm^3^ (corresponding to a matrix size of 128 × 104 pixels) during the free breathing period. The other imaging parameters were a slice thickness of 5 mm, an echo time (TE) of 59 ms, a repetition time (TR) of 2050 ms, a GRAPPA parallel imaging factor of 2, an MEG frequency of 43.48 Hz for 30, 40, and 50 Hz vibrations and 44.88 Hz for 60 Hz vibration, and an MEG amplitude of 30 mT/m. The total time for the MRE measurements was approximately 3.5 min. Technical success of tomoelastography was evaluated by visual assessment of the shear wave images by an experienced operator.

Multifrequency wave field data were processed using the processing pipeline [19]. High spatial resolution maps of shear wave velocity (c) and phase angle (*φ*) over the entire FOV were obtained. Since *c* is proportional to the square root of the storage modulus, while *φ* continuously changes between 0 (solid properties) and π/2 (liquid properties), it can be considered to represent stiffness and fluidity, respectively.

### 2.4. Histopathological Analysis

Histopathological analysis was performed by a pathologist with 10 years of experience in liver pathology, who was blinded to the radiological and clinical findings. Immunohistochemistry staining for Ki-67 protein was performed. The Ki-67 expression was assessed by noting the percentage of positively stained cells with brownish yellow nuclei. Then, HCC was classified as low Ki-67 expression (Ki-67 ≤ 20%) and high Ki-67 expression (Ki-67 > 20%) groups.

### 2.5. Image Preprocessing

DCE and DWI images were registered to T2-weighted image using ANTs toolbox [20]. The B-spline interpolation was applied to the conventional MR images to normalize the image resolution to 0.9 × 0.9 × 4.0 mm^3^ and FOV to 360 × 360 mm^2^. The tumor region of interest (ROI) was manually annotated by an abdominal radiologist (with five years of experience) on T2-weighted images, *c*-map, and *φ*-map using ITK-SNAP software [21].

### 2.6. cMRI-Based DLCR Model

Six CNN network architectures (Inception-Resnet [22], Inception [23], Resnet [24], VGG16 [25], VGG19 [26], and Xception [26]) were used and compared for the extraction of deep and high-dimensional image features based on T2WI, DWI, and DCE images (Appendix A). In addition, the radiomics features were extracted from the same modalities using the PyRadiomics toolbox in Python [27], which included first order features, shape features, gray level co-occurrence matrix (GLCM) features, gray level size zone matrix (GLSZM) features, gray level run length matrix (GLRLM) features, neighboring gray tone difference matrix (NGTDM) features, and gray level dependence matrix (GLDM) features.

Clinical data were also fed into the predictive model, including alpha-fetoprotein (AFP), platelet count, prealbumin level, alanine aminotransferase (ALT) level, aminotransferase aspartate (AST) level, total bilirubin, direct bilirubin, albumin level, etc. To avoid overfitting, LASSO regression (λ = 0.001) with 5-fold cross-validation was applied to reduce the dimensionality of features. Finally, the top six features were filtered out for further modeling. After features reduction, an SVM model (with penalty parameter = 1.0, kernel = ‘rbf’, order of polynomial kernel function = 3) was employed for the final prediction of high Ki-67 expression. The different network architectures (i.e., Inception-Resnet, Inception, Resnet, VGG16, VGG19, and Xception) were compared in the DLCR model based on conventional MRI (including T2, DWI, and DCE). After comparison, the best model was determined (Figure 2).

### 2.7. cMRI-Based DLCR Model with Tomoelastography

The cMRI-based DLCR model incorporated traditional MRI modalities. On this basis, both the *c*-map and *φ*-map were integrated to establish the model to compare prediction effects. To visualize the contributions of each feature, a random forest-based feature ranking was implemented (Figure 3).

### 2.8. Statistical Analysis

Continuous data were expressed as mean ± standard deviation. Categorical data were expressed as number and percentages. Receiver operating characteristic (ROC) analysis was conducted to evaluate the performance of two models in predicting high Ki-67 expression. Comparisons between the AUCs were conducted using the Delong test. A two-tailed Student’s *t*-test with a *p*-value of <0.05 was considered statistically significant.

## 3. Results

### 3.1. Demographics and Clinical Characteristics

In the training cohort, the high Ki-67 expression group included 40 patients while the low Ki-67 expression group included 47 patients. In the testing cohort, the high Ki-67 expression group included 9 patients while the low Ki-67 expression group included 12 patients. In the independent testing cohort, the high Ki-67 expression group included 17 patients while low Ki-67 expression group included 26 patients. Clinical characteristics were not significantly different between the training and validation cohorts. The detailed information is summarized in Table 2.

Among all 108 patients in the internal cohort, high Ki-67 expression was pathologically diagnosed in 49 patients (45.4%). Compared to the low Ki-67 expression groups, the high Ki-67 expression group had elevated AFP levels (49.35 vs. 277.17 mg/mL). *p* value (bolded) represented significant difference between the two groups. The other demographic and laboratory parameters showed no significant difference between the two groups.

### 3.2. Optimization of cMRI-Based DLCR Models

When comparing the CNN models based on the six network architectures, the Xception architecture showed the best performance for classifying high Ki-67 expression, with an AUC of 0.80 ± 0.03 (95% confidence interval (CI): 0.79–0.81) and an accuracy of 0.77 ± 0.04 (95%CI: 0.76–0.78). The predictive power of the other five network architectures were as follows: Inception-Resnet (AUC = 0.71 ± 0.04 (95%CI: 0.70–0.72)), Resnet (AUC = 0.70 ± 0.04 (95%CI: 0.69–0.70)), Inception (AUC = 0.65 ± 0.03 (95%CI:0.64–0.66)), VGG19 (AUC = 0.65 ± 0.03 (95%CI: 0.64–0.66)), and VGG16 (AUC = 0.62 ± 0.03 (95%CI: 0.61–0.63)) (Table 3). Thus, the Xception architecture was selected as the final deep learning feature extraction. By adding clinical data as input, the AUC achieved to 0.84 ± 0.03 (95%CI: 0.83–0.85) in the validation cohort and 0.74 ± 0.02 (95%CI: 0.73–0.75) in the independent testing cohort.

### 3.3. Comparison of DLCR Models with/without Tomoelastography

Based on the training cohort, both the cMRI-based DLCR model and the DLCR model with tomoelastography were established. When combined tomoelastography with cMRI and clinical data, the model achieved a higher AUC of 0.90 ± 0.03 (95%CI: 0.89–0.91) than the AUC of 0.84 ± 0.03 (95%CI: 0.83–0.85) for the DLCR model without tomoelastography in the validation cohort.

For the independent testing cohort, the DLCR model with tomoelastography also performed better than the cMRI-based DLCR model. The AUC of the cMRI-based DLCR model was 0.74 ± 0.02 (95%CI: 0.73–0.75). Accuracy was just 0.72 ± 0.03 (95%CI: 0.71–0.73). The AUC achieved 0.83 ± 0.03 (95%CI: 0.82–0.84) and the corresponding accuracy was 0.83 ± 0.02 (95%CI: 0.82–0.84) for the DLCR model with tomoelastography (Table 4).

### 3.4. Contribution of Predictive Efficacy

Both *c* and *φ* values were ranked within the top six features for Ki-67 prediction with random forest selection (Figure 3b). The top two features were related to the viscosity parameter *φ*, which revealed the value of MRE-based viscosity for the assessment of the tumor proliferation status in HCC. Two representative cases with different Ki-67 levels are shown in Figure 4. The lesions with higher Ki-67 expression tended to have higher *φ*.

## 4. Discussion

Our study demonstrated that the MRE-based viscoelasticity map improved the performance of the DLCR model for Ki-67 expression in HCC. The results suggested that biomechanical properties, elasticity, and viscosity could provide additional information for the assessment of the HCC proliferation status. To the best of our knowledge, this is the first study to investigate contribution of biomechanics for HCC aggressiveness using machine learning methods.

According to the AUC, our model achieved better performance than the previously mentioned studies, and the reason can be explained because more imaging biomarkers were introduced in the predictive model. Our model used both deep learning and radiomics, and innovatively incorporated MRE. Compared to the models that only included conventional MRI and routine clinical data for the prediction of Ki-67 expression, the proposed DLCR model combined cMRI as well as MRE provided more information of the tumor characteristics. The tomoelastography that we used in this study provided high-resolution parameter maps, which can reflect the mechanical properties of the tissue accurately instead of only the total water content in the conventional MRI [28].

It is worth noting that several studies have found the value of elasticity in predicting the Ki-67 expression of breast cancer. Cha et al. [29] had found that higher elasticity value was associated with Ki-67 and the invasive size of the tumor in breast cancer. Choi et al. [30] discovered that Ki-67 positivity, high nuclear grade, large tumor (invasive) size, and so on, were associated with a significantly high shear wave elasticity ratio. However, its application in the prediction of HCC Ki-67 expression has not yet been investigated. In addition, those studies of breast cancer only measured the shear wave velocity by ultrasound, while we used two-dimensional tomoelastography in the DLCR model, which can better capture the textures for modeling and provide another vital biomechanical parameter *φ*, representing the viscosity of tissue.

MRE quantified the stiffness of tissues with structural and elastic characteristics, which was widely reported for tumor detection and classification [31,32]. However, the previous studies only reported the simple relationship between tumor aggressiveness and MRE-based measurements, while we used *c* and *φ* maps as inputs of the DLCR model in order to get more detailed and spatial information about the tumor biomechanical properties. Our results showed that tomoelastography images, especially the *φ*-map, improved the performance of the DLCR model significantly. The features extracted by CNN and the zone variance value extracted by radiomics from the *φ*-map ranked the top two contributions in the DLCR model for the prediction task. In other words, the viscosity of HCC, which represents the fluidity properties, showed a good predictive ability of tumor aggressiveness.

The elevated fluidity was supposed to be associated with higher vessel density and the increased intracellular fluid mobility of tumor cells [30]. Higher Ki-67 expression is associated with faster progression and poorer prognosis for HCC patients. During tumor progression, increased vascularity and cellularity, a denser extracellular matrix, and higher interstitial pressure influenced the tumor stiffness and fluidity [33,34]. Moreover, constant elevating blood supply for maintaining tumor growth and development also had an impact on the tumor fluidity. Similarly, previous studies had found that the fluidity of tumors, which was related to angiogenesis and intercellular friction, have a significant contribution to the development and proliferation of malignant tumors [12,30]. In addition, solid tumors become aggressive by metastatic spread, which requires partial fluidification so that cancer cells can move. These findings show the close relationship between tumor fluidity and aggressiveness and can help to explain why the features from the *φ*-map ranked the top two contributions in the DLCR model.

Nevertheless, the present study had some limitations. First, the sample size was not large enough. Although we tested our model on an independent cohort, more validations are still needed, especially multicenter validations. Second, there is currently no standardized Ki-67 expression level threshold in HCC, which varies from 10% to 30% in studies. We treated the Ki-67 prediction task as a binary classification problem (with a threshold of 20%) according to previous studies [35,36] in order to make sure that the two groups had roughly the same number of people. A better solution might be dividing the Ki-67 expression into several subgroups. Third, the use of deep learning models was more targeted, which may not have widespread application. The complexity of the Inception architecture made it harder to make changes to the network. Xception performed better on the small dataset. As the number of layers increased, the VGG network showed a degradation problem. We only compared six widely used network architectures for the prediction task. More recently proposed architectures such as lightBGM can be used to further improve the prediction performance. Fourth, the morphological features from conventional MRIs, such as the LI-RADS classification [37] and other pathological features like MVI, are very important in evaluating the HCC aggressiveness and the patients’ prognosis. Further work about the other prognostic factors for the HCC prognosis from the point of view of the biomechanical can be done.

## 5. Conclusions

In conclusion, our study proposed a noninvasive DLCR model and explored the added value of multifrequency MRE in the prediction of Ki-67 expression in HCC patients. The experimental results proved that the shear wave speed and phase angle improved the performance of the DLCR model significantly, suggesting that MRE-based *c* and *φ*-maps can serve as important parameters to assess tumor proliferation status in HCC.

## Figures and Tables

**Figure 1 cancers-14-02575-f001:**
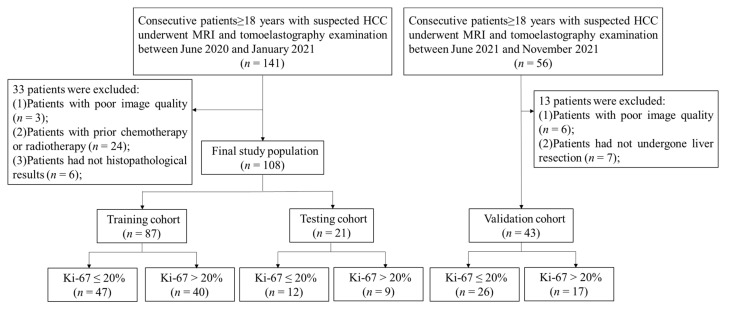
Flowchart of inclusions and exclusion criteria of participants.

**Figure 2 cancers-14-02575-f002:**
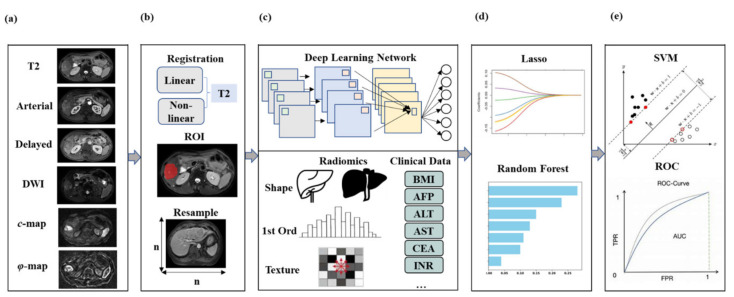
Diagram of the DLCR model, including procedures of (**a**) image selection, (**b**) image preprocessing, (**c**) feature extraction, (**d**) feature reduction, and (**e**) prediction of Ki-67 expression.

**Figure 3 cancers-14-02575-f003:**
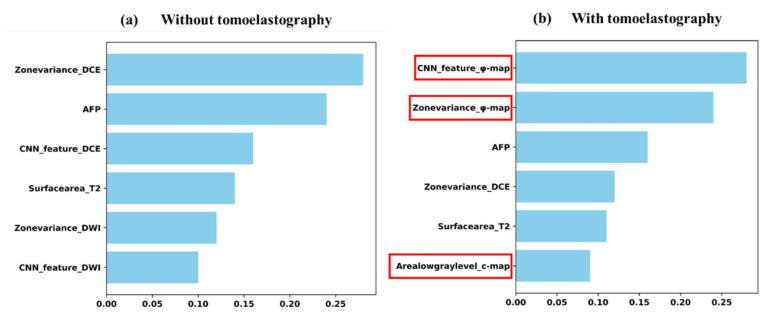
Weighting and ranking of the top six predominant features for the prediction of Ki-67 expression calculated by random forest (**a**) without and (**b**) with the inclusion of tomoelastography-derived *c* and *φ* maps. With the introduction of tomoelastography, two were related to the *φ*-map, which were the top 2 in importance, and one was related to the c-map, which demonstrated the importance of features extracted by c-map and *φ*-map (red squares) in Ki-67 expression prediction.

**Figure 4 cancers-14-02575-f004:**
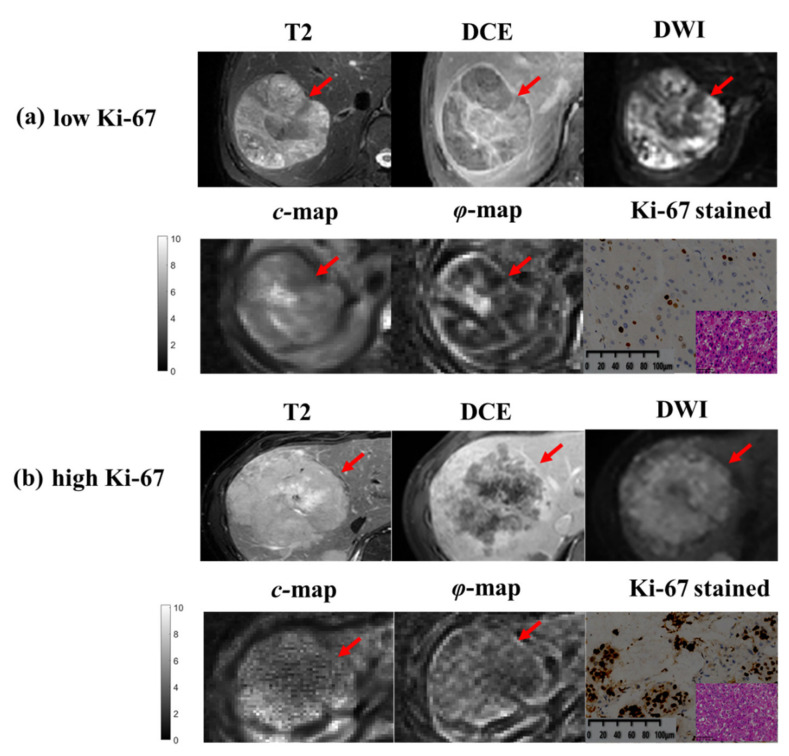
Representative MRI, tomoelastography parameter maps and Ki-67 stained images (magnification, ×400, corresponding HE stained images at bottom right) of HCCs that have (**a**) low (Ki-67 = 10%) and (**b**) high (Ki-67 = 60%) Ki-67 expression. The two HCC lesions (red arrow) showed similar imaging patterns on the conventional MRI while different imaging modes on tomoelastography.

**Table 1 cancers-14-02575-t001:** Demographics and clinical characteristics of the participants in this study.

Variable	Total (*n* = 108)	Training (*n* = 87)	Testing (*n* = 21)	*p*-Value
Age (years)	59.57 ± 10.97	59.38 ± 11.13	60.38 ± 10.20	0.19
Sex, *n* (%)	93 (87.03%)	76 (88.51%)	17 (80.95%)	0.83
BMI (kg/m^2^)	23.81 ± 3.01	23.63 ± 2.94	24.59 ± 3.15	0.45
Etiology, No.	--	-	-	<0.05
Hepatitis B virus	83	64	18	-
Hepatitis C virus	4	3	1	-
Others	21	19	2	-
AFP level (mg/mL)	-	-	-	<0.05
<20	50	40	10	-
≥20	58	47	11	-
Platelet count (×10^9^/L)	143.06 ± 64.93	141.81 ± 65.41	148.40 ± 62.56	0.39
Prealbumin level (mg/L)	195.99 ± 60.55	195.74 ± 62.60	197.05 ± 50.78	0.31
ALT level (IU/L)	41.08 ± 60.21	41.23 ± 65.27	40.45 ± 29.96	0.27
AST level (IU/L)	45.25 ± 64.87	46.37 ± 71.14	40.45 ± 22.68	0.25
Total bilirubin (μmol/L)	18.95 ± 12.09	18.52 ± 8.02	20.77 ± 22.23	0.16
Direct bilirubin (μmol/L)	3.89 ± 2.96	3.90 ± 2.72	3.83 ± 3.84	0.35
Albumin level (g/L)	34.77 ± 11.81	39.86 ± 5.87	40.30 ± 7.70	0.27
Prothrombin time (s)	12.54 ± 1.13	12.14 ± 1.36	12.89 ± 0.91	0.44
INR	1.04 ± 0.12	1.03 ± 0.12	1.10 ± 0.08	0.57
Ki-67(%)	27.28 ± 20.47	27.55 ± 19.52	26.14 ± 23.98	0.26

**Table 2 cancers-14-02575-t002:** Comparison of the demographics and clinical characteristics between high Ki-67 expression groups and low Ki-67 expression groups.

Variable	Training (*n* = 87)		*p* Value	Validation (*n* = 21)		*p* Value	Testing (*n* = 43)		*p* Value
	High Ki-67 (*n* = 40)	Low Ki-67 (*n* = 47)		High Ki-67 (*n* = 9)	Low Ki-67 (*n* = 12)		High Ki-67 (*n* = 17)	Low Ki-67 (*n* = 26)	
Age (years)	56.8 ± 11.5	65.0 ± 7.9	0.07	61.8 ± 9.5	59.3 ± 10.6	0.09	59.4 ± 11.7	60.7 ± 10.5	0.17
Sex, *n* (%)	35 (87.50%)	41 (87.23%)	0.78	7 (77.78%)	10 (83.33%)	0.81	15 (88.24%)	21 (80.77%)	0.38
BMI (kg/m2)	23.30 ± 2.83	24.32 ± 3.05	0.57	26.02 ± 2.46	23.51 ± 3.19	0.67	23.18 ± 2.34	25.35 ± 4.06	0.27
Etiology, No.									
Hepatitis B virus	27 (67.50%)	39 (82.98%)		8 (88.89%)	10 (83.33%)		11 (64.71%)	18 (69.23%)	
Hepatitis C virus	3 (7.50%)	1 (2.13%)		1 (11.11%)	1 (8.33%)		2 (11.76%)	1 (3.85%)	
Others	10 (25.00%)	7 (14.89%)		0 (0%)	1 (8.33%)		4 (23.53%)	7 (26.92%)	
AFP level (mg/mL)			**0.03**			**0.02**			**0.04**
<20	6 (15.00%)	38 (80.85%)		3 (33.33%)	8 (66.67%)		4 (23.53%)	15 (57.69%)	
≥20	34 (87.50%)	9 (19.15%)		6 (66.67%)	4 (33.33%)		13 (76.47%)	11 (42.31%)	
Platelet count (×10^9^/L)	142.32 ± 70.08	140.70 ± 53.78	0.35	133.50 ± 40.53	158.33 ± 71.99	0.34	156.00 ± 89.57	152.32 ± 73.11	0.36
Prealbumin level (mg/L)	187.86 ± 54.95	212.96 ± 73.84	0.24	194.13 ± 45.34	199.00 ± 54.01	0.31	177.18 ± 58.22	178.44 ± 53.98	0.27
ALT level (IU/L)	45.32 ± 77.39	32.30 ± 19.06	0.35	53.25 ± 36.98	31.92 ± 20.05	0.34	35.24 ± 22.25	33.40 ± 17.74	0.45
AST level (IU/L)	50.76 ± 84.47	36.78 ± 19.83	0.15	46.63 ± 27.50	36.33 ± 17.63	0.12	49.53 ± 41.14	36.80 ± 13.22	0.12
Total bilirubin (μmol/L)	18.23 ± 7.51	19.18 ± 9.00	0.36	15.35 ± 4.38	24.38 ± 27.90	0.17	17.09 ± 5.09	16.54 ± 7.70	0.24
Direct bilirubin (μmol/L)	3.99 ± 2.90	3.70 ± 2.26	0.26	3.00 ± 1.04	4.38 ± 4.81	0.28	3.79 ± 2.18	3.53 ± 2.24	0.23
Albumin level (g/L)	39.58 ± 4.68	40.48 ± 7.83	0.39	37.75 ± 2.90	42.00 ± 9.27	0.41	38.65 ± 4.73	38.92 ± 4.07	0.72
Prothrombin time (s)	12.21 ± 1.33	12.01 ± 1.39	0.81	12.66 ± 0.52	13.04 ± 1.07	0.89	12.47 ± 0.76	12.56 ± 1.11	0.82
INR	1.04 ± 0.12	1.02 ± 0.12	0.67	1.08 ± 0.05	1.11 ± 0.10	0.57	1.06 ± 0.07	1.07 ± 0.10	0.67
*c* (rad)	2.45 ± 0.65	2.26 ± 0.66	0.17	2.38 ± 0.85	2.23 ± 0.97	**0.05**	2.07 ± 0.58	2.11 ± 0.61	0.11
*φ* (m/s)	1.14 ± 0.25	1.05 ± 0.24	0.09	1.20 ± 0.24	0.99 ± 0.20	0.71	1.03 ± 0.22	1.02 ± 0.25	0.20

*p* value (bolded) represented significant difference between the two groups.

**Table 3 cancers-14-02575-t003:** Comparison of different CNN model architectures in the prediction task.

Model	Inception-Resnet		Xception		Inception		Resnet		VGG16		VGG19	
AUC	0.71 ± 0.04	0.61 ± 0.03	**0.80 ± 0.03**	0.71 ± 0.02	0.65 ± 0.03	0.56 ± 0.03	0.70 ± 0.04	0.62 ± 0.03	0.62 ± 0.03	0.53 ± 0.03	0.65 ± 0.03	0.55 ± 0.05
	(0.70–0.72)	(0.60–0.62)	**(0.79–0.81)**	(0.70–0.72)	(0.64–0.66)	(0.55–0.57)	(0.69–0.71)	(0.61–0.63)	(0.61–0.63)	(0.52–0.54)	(0.64–0.66)	(0.54–0.57)
Accuracy	0.71 ± 0.05	0.61 ± 0.04	**0.77 ± 0.04**	0.68 ± 0.03	0.66 ± 0.05	0.57 ± 0.04	0.70 ± 0.04	0.61 ± 0.04	0.62 ± 0.04	0.53 ± 0.03	0.64 ± 0.04	0.55 ± 0.03
	(0.70–0.72)	(0.60–0.62)	**(0.76–0.78)**	(0.67–0.69)	(0.65–0.67)	(0.56–0.58)	(0.69–0.71)	(0.60–0.62)	(0.61–0.63)	(0.52–0.54)	(0.63–0.65)	(0.54–0.56)
Sensitivity	0.68 ± 0.05	0.60 ± 0.03	**0.76 ± 0.06**	0.67 ± 0.04	0.65 ± 0.04	0.57 ± 0.05	0.67 ± 0.05	0.59 ± 0.03	0.59 ± 0.03	0.53 ± 0.04	0.66 ± 0.04	0.57 ± 0.02
	(0.67–0.69)	(0.59–0.61)	**(0.75–0.77)**	(0.66–0.68)	(0.65–0.67)	(0.55–0.58)	(0.66–0.68)	(0.58–0.60)	(0.58–0.60)	(0.52–0.54)	(0.65–0.67)	(0.56–0.58)
Specificity	0.72 ± 0.04	0.63 ± 0.02	**0.78 ± 0.06**	0.68 ± 0.04	0.67 ± 0.05	0.58 ± 0.04	0.72 ± 0.03	0.58 ± 0.04	0.64 ± 0.04	0.55 ± 0.03	0.62 ± 0.04	0.52 ± 0.03
	(0.71–0.73)	(0.62–0.64)	**(0.77–0.79)**	(0.67–0.69)	(0.66–0.68)	(0.57–0.59)	(0.71–0.73)	(0.57–0.59)	(0.63–0.65)	(0.54–0.56)	(0.61–0.63)	(0.51–0.53)
PPV	0.69 ± 0.03	0.63 ± 0.02	**0.76 ± 0.03**	0.65 ± 0.04	0.64 ± 0.02	0.55 ± 0.04	0.67 ± 0.02	0.58 ± 0.02	0.59 ± 0.04	0.52 ± 0.03	0.65 ± 0.03	0.56 ± 0.04
	(0.68–0.70)	(0.62–0.64)	**(0.75–0.77)**	(0.64–0.66)	(0.64–0.65)	(0.54–0.56)	(0.67–0.68)	(0.57–0.59)	(0.58–0.60)	(0.51–0.53)	(0.64–0.66)	(0.55–0.57)
NPV	0.71 ± 0.02	0.62 ± 0.03	**0.77 ± 0.04**	0.68 ± 0.03	0.68 ± 0.03	0.59 ± 0.04	0.72 ± 0.01	0.55 ± 0.03	0.64 ± 0.03	0.55 ± 0.02	0.63 ± 0.02	0.54 ± 0.03
	(0.71–0.72)	(0.62–0.64)	**(0.76–0.78)**	(0.67–0.69)	(0.67–0.69)	(0.58–0.60)	(0.72–0.73)	(0.54–0.56)	(0.63–0.65)	(0.54–0.56)	(0.63–0.64)	(0.53–0.55)

AUC = area under curve, PPV = positive predictive value, NPV = negative predictive value. Xception performed best among all six models (bolded).

**Table 4 cancers-14-02575-t004:** Performance of the DLCR models with different combinations of parameters.

Cohort	Parameter Combinations	Evaluation					
		AUC	Accuracy	Sensitivity	Specificity	PPV	NPV
Internal validation cohort	cMRI + AFP	0.84 ± 0.03	0.81 ± 0.04	0.80 ± 0.06	0.82 ± 0.06	0.78 ± 0.06	0.80 ± 0.03
		(0.83–0.85)	(0.80–0.82)	(0.79–0.81)	(0.81–0.83)	(0.77–0.79)	(0.79–0.81)
	cMRI + AFP + MRE	0.90 ± 0.03	0.87 ± 0.05	0.86 ± 0.04	0.93 ± 0.02	0.84 ± 0.03	0.87 ± 0.02
		(0.89–0.91)	(0.86–0.88)	(0.85–0.87)	(0.93–0.94)	(0.83–0.85)	(0.87–0.88)
Independent testing cohort	cMRI + AFP	0.74 ± 0.02	0.72 ± 0.03	0.72 ± 0.05	0.72 ± 0.04	0.68 ± 0.05	0.71 ± 0.03
		(0.73–0.75)	(0.71–0.73)	(0.71–0.74)	(0.71–0.73)	(0.67–0.70)	(0.70–0.72)
	cMRI + AFP + MRE	0.83 ± 0.03	0.83 ± 0.02	0.80 ± 0.03	0.86 ± 0.01	0.78 ± 0.02	0.80 ± 0.03
		(0.82–0.84)	(0.82–0.84)	(0.79–0.81)	(0.86–0.87)	(0.77–0.79)	(0.79–0.81)

cMRI = conventional magnetic resonance imaging (including T2 + DWI + DCE); AFP = α-fetoprotein; DCE = dynamic contrast enhanced; MRE = magnetic resonance elastography.

## Data Availability

Data presented in this study are available, by request, from the corresponding authors, due to matters of privacy.

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
