# Peer review of "Added Value of Viscoelasticity for MRI-Based Prediction of Ki-67 Expression of Hepatocellular Carcinoma Using a Deep Learning Combined Radiomics (DLCR) Model"

_cancers, 2022, doi:10.3390/cancers14112575_

Round 1

Reviewer 1 Report

This interesting paper from Hu et al entitled “Added value of viscoelasticity for MRI-based prediction of Ki-67 expression of hepatocellular carcinoma using a deep learning combined radiomics (DLCR) model” provides an interesting application of AI on MRI-based HCC diagnosis

The implementation of imaging techniques coupled by development of machine learning has triggered novel approach to cancer medicine, including HCC

The subject is original, but I have a couple of suggestions for improve the paper:

1) it is fundamental to discuss the role of other progrnostic factors for HCC prognosis such as MVI, satellitosis or capsule inifiltratiion, referring to recently published literature (10.3390/diagnostics12010160)

2) the Authors should better acklowledge the current limitations concerning widespread application of deep learning models

3) include the lack of pathology data oin studuy limitation

Best regards

Reviewer 2 Report

I appreciate you reading an interesting article, which shows that CNN-supporting tomoelastography can predict the ki-67 labelling index of hepatocellular carcinoma. Here are my recommendations for you.

  1. What makes the remarkable differences with/without tomoelastography in Figure 3?
  2. What is the reference to dividing low/high expression of Ki-67 based on 20%? 
  3. Is there any possibility to overestimate the ki-67 LI of HCC? HCC often shows inflammatory cell infiltration, and immune cells show high ki-67 LI. How did you exclude immune cells ki-67 stained in HCC only based on tomoelastography?
  4. Is tomoelastography commonly used in HCC?
  5. How about showing me some H&E and Ki-67 stained HCC slide images of representative cases you want to?

Reviewer 3 Report

Dear Authors,

In a retrospective study 108 patients were included with histopathology-proven hepatocellular carcinoma and in a second time, a validation cohort included 43 patients.  The study goal was to investigate the interest to include maps derived from MR elastography into a deep learning combined radiomics model.  Models with MRE as input showed the highest AUC. Overall, the study is well written and organized.

Congratulations to the authors, it is a well-done study, showing the interest of MR elastography parameters in radiomics models. However, there are a number of issues to be addressed:

  1. Although the main message here is about the DLCR model and maps derived from MRE as inputs. It will be interesting to know the ability of shear wave speed and phase angle to diagnose high Ki-67 level. As it is mentioned by the authors in paragraph 3.4 the lesions with high Ki-67 expression tended to have higher phase angle. Then, does phase angle alone can detect patients with high Ki-67 level?

Minors,

  1. Page 3, line 102. It will be interesting to add to references 9 and 10 a study on MRE applied on liver or HCC as the present work is about HCC.

  1. Page 3, line 109. The elastic modulus is not a physical parameter. I assume that the complex shear modulus. Can you confirm and correct the name of this parameter?

  1. Page 3, line 116. Why the term ‘et al.’ is used. According to the references only remain liver cancer. Can you replace ‘et al’ by liver cancer.

  1. Page 3, line 119. The reference 17 is ‘classic MR elastography’ applied on small animals and does not match with this paragraph which is about tomoelastography.

  1. Table 1. The table about the demographics and clinical characteristics is great, however, would it be possible to add p values for etiology and AFP level. I think it is possible with a chi-squared test.

  1. Figure 4. Does the c-map and phi-map are at the same location? They are derived from the same acquisition so they should be, nevertheless, they seem to be different for the patient b.

  1. Page 12, line 407. The reference 14 is incorrectly formatted.

Reviewer 4 Report

This paper demonstrated the possible added value of viscoelasticity measured by magnetic resonance elastography (MRE), which means images of shear wave speed (c-map) and phase angle (φ-map), in the prediction of Ki-67 expression in HCC using a deep learning combined radiomics (DLCR) model.

Authors clearly showed that the model with all modalities (MRE, AFP, and cMRI=T2WI, DCE, and DWI) as inputs achieved the highest AUC and was validated in internal and external validation cohorts.

Apparently, MRE-based c and φ-maps could serve as important parameters to assess tumor proliferation activity in HCC on the basis of data presented in this manuscript.

Basically, the authors are to be commended for their well-thought-out and detailed study based on sound procedure. Although it may be a biased substantially small study in a single institution, it is so thought-provoking that there are no other speculations to compare to at this time, and I am willing to admit this biomechanical model.

However, the impact of the information presented here is somewhat weak because we cannot directly consider that six ranked features by random forest after filtering-out with LASSO regression do mean elasticity or viscosity (corresponding to MRE-based c and φ-maps) with inclusion of tomoelastography in Fig. 3.

Indeed, I am concerned that even the fact of their novelty of suggesting proposed biomechanical properties, elasticity and viscosity, could provide additional information for assessment of HCC proliferation status, I do not see c values in the Fig. 3 (b). This causes the entire consideration to be partially flawed.

In addition, elasticity related to tumor aggressiveness is emphasized in the Discussion thorough their findings, deep insights regarding viscosity seemingly disappeared from argument. Authors may want to explain this point clearly.

And as pointed out in the text, Ki-67 threshold was set to 20% for simplicity.  The authors need to show, with appropriate citations, that adoption of this threshold would clearly divide the prognosis for this disease.

Minor comment:

In 2.1. Patients, line 142, ‘a ratio of 7:3’ is mistaking of ‘8:2’.

Round 2

Reviewer 1 Report

The Authors provided a valuable revision of their original manuscript

The paper can be accepted for publication

Best regards

Reviewer 2 Report

Thank you for revision, which makes yours more novel and interesting.